# Oxidative Stress in Type 2 Diabetes: Impacts from Pathogenesis to Lifestyle Modifications

Alfredo Caturano [1,2,*,†], Margherita D'Angelo [2,3,†], Andrea Mormone [1], Vincenzo Russo [3,4],
Maria Pina Mollica [5], Teresa Salvatore [6], Raffaele Galiero [1], Luca Rinaldi [1], Erica Vetrano [1],
Raffaele Marfella [1], Marcellino Monda [2], Antonio Giordano [3,‡] and Ferdinando Carlo Sasso [1,‡]

1   Department of Advanced Medical and Surgical Sciences, University of Campania Luigi Vanvitelli,
    I-80138 Naples, Italy
2   Department of Experimental Medicine, University of Campania Luigi Vanvitelli, I-80138 Naples, Italy;
    meghyd@icloud.com (M.D.)
3   Department of Biology, Sbarro Institute for Cancer Research and Molecular Medicine, College of Science and
    Technology, Temple University, Philadelphia, PA 19122, USA
4   Division of Cardiology, Department of Medical Translational Sciences, University of Campania Luigi
    Vanvitelli, I-80138 Naples, Italy
5   Department of Biology, University of Naples Federico II, I-80134 Naples, Italy
6   Department of Precision Medicine, University of Campania Luigi Vanvitelli, I-80138 Naples, Italy
*   Correspondence: alfredo.caturano@unicampania.it
†   These authors contributed equally to this work.
‡   These authors contributed equally to this work.

**Abstract:** Oxidative stress is a critical factor in the pathogenesis and progression of diabetes and its associated complications. The imbalance between reactive oxygen species (ROS) production and the body's antioxidant defence mechanisms leads to cellular damage and dysfunction. In diabetes, chronic hyperglycaemia and mitochondrial dysfunction contribute to increased ROS production, further exacerbating oxidative stress. This oxidative burden adversely affects various aspects of diabetes, including impaired beta-cell function and insulin resistance, leading to disrupted glucose regulation. Additionally, oxidative stress-induced damage to blood vessels and impaired endothelial function contribute to the development of diabetic vascular complications such as retinopathy, nephropathy, and cardiovascular diseases. Moreover, organs and tissues throughout the body, including the kidneys, nerves, and eyes, are vulnerable to oxidative stress, resulting in diabetic nephropathy, neuropathy, and retinopathy. Strategies to mitigate oxidative stress in diabetes include antioxidant therapy, lifestyle modifications, and effective management of hyperglycaemia. However, further research is necessary to comprehensively understand the underlying mechanisms of oxidative stress in diabetes and to evaluate the efficacy of antioxidant interventions in preventing and treating diabetic complications. By addressing oxidative stress, it might be possible to alleviate the burden of diabetes and improve patient outcomes.

**Keywords:** oxidative stress; type 2 diabetes; diet; Mediterranean diet; physical activity; lifestyle modifications; diabetes complications

## 1. Introduction

Oxidative stress has garnered significant attention in recent years due to its profound impact on human health, particularly its association with diabetes. Oxidative stress is a condition that occurs when there is an imbalance between the production of reactive oxygen species (ROS) and the body's antioxidant defence system's ability to neutralise them. ROS, including free radicals and other highly reactive molecules, are natural byproducts of normal cellular metabolism. However, when their production exceeds the capacity of the body's antioxidants to counteract them, oxidative stress occurs [1,2].

This imbalance leads to a cascade of detrimental effects within the body. The excessive accumulation of ROS can damage various cellular components, including proteins, lipids, and DNA, resulting in cellular dysfunction and disruption of normal physiological processes. Such damage can trigger inflammation and impair the function of vital cellular structures, ultimately contributing to the development and progression of various diseases, including diabetes [3].

Diabetes is a chronic disease characterised by elevated blood glucose levels resulting from either an absolute or relative lack of insulin or impaired insulin functionality. Diabetes mellitus poses a significant challenge in healthcare, being one of the leading health issues worldwide. Its increasing prevalence can be primarily attributed to factors such as an ageing population, socio-economic development, unhealthy dietary patterns, and sedentary lifestyles. In fact, as of 2019, it was estimated that 9.3% of the global adult population (20–79 years) lived with diabetes, and this figure is estimated to rise in the near future [4,5]. The ageing process encompasses a multifaceted biological decline, including reduced physical capabilities. A pivotal contributor to this ageing progression is mitochondrial dysfunction. Mitochondria, responsible for energy production, also internally generate ROS. As people age, ATP production diminishes, while ROS production increases concurrently with a weakening of antioxidant defences [6]. The persistent high blood sugar levels characteristic of diabetes foster heightened ROS production through mechanisms such as glucose auto-oxidation, activation of the polyol pathway, and the development of advanced glycation end products (AGEs). Additionally, in diabetes, the prevalent mitochondrial dysfunction results in an added release of ROS [7].

The consequences of oxidative stress in diabetes are widespread and multifaceted. Beta-cell dysfunction and apoptosis occur due to the damaging effects of ROS, further compromising insulin production and secretion. Oxidative stress also contributes to insulin resistance, impairing the ability of insulin to facilitate glucose uptake into cells [8].

Moreover, the deleterious effects of oxidative stress extend to the vascular system. Increased ROS production damages blood vessels, promotes inflammation, and impairs endothelial function. These factors contribute to the development of diabetic vascular complications, including diabetic retinopathy (a leading cause of blindness), nephropathy, other sensorial damage, and cardiovascular diseases [8–10].

Furthermore, oxidative stress-induced damage affects various organs and tissues, exacerbating diabetic complications. Kidneys, nerves, and eyes are particularly vulnerable. Oxidative stress disrupts normal renal function, leading to diabetic nephropathy. It also damages nerves, contributing to the development of diabetic neuropathy, which can manifest as pain, numbness, and impaired sensations. Additionally, oxidative stress plays a role in the development of diabetic retinopathy, leading to vision impairment or even blindness [11,12].

In this review, we aim to emphasise the existing evidence that demonstrates the strong association between type 2 diabetes mellitus (T2DM) and oxidative stress, along with the progression of complications related to diabetes.

## 2. Genesis of Oxidative Stress in Diabetes

Oxidative stress in diabetes arises from a complex interplay of various factors, including the accumulation of glycolysis intermediates, activation of the polyol pathway, formation of AGEs, activation of Protein Kinase C (PKC), and activation of the hexosamine pathway (Figure 1) [13].

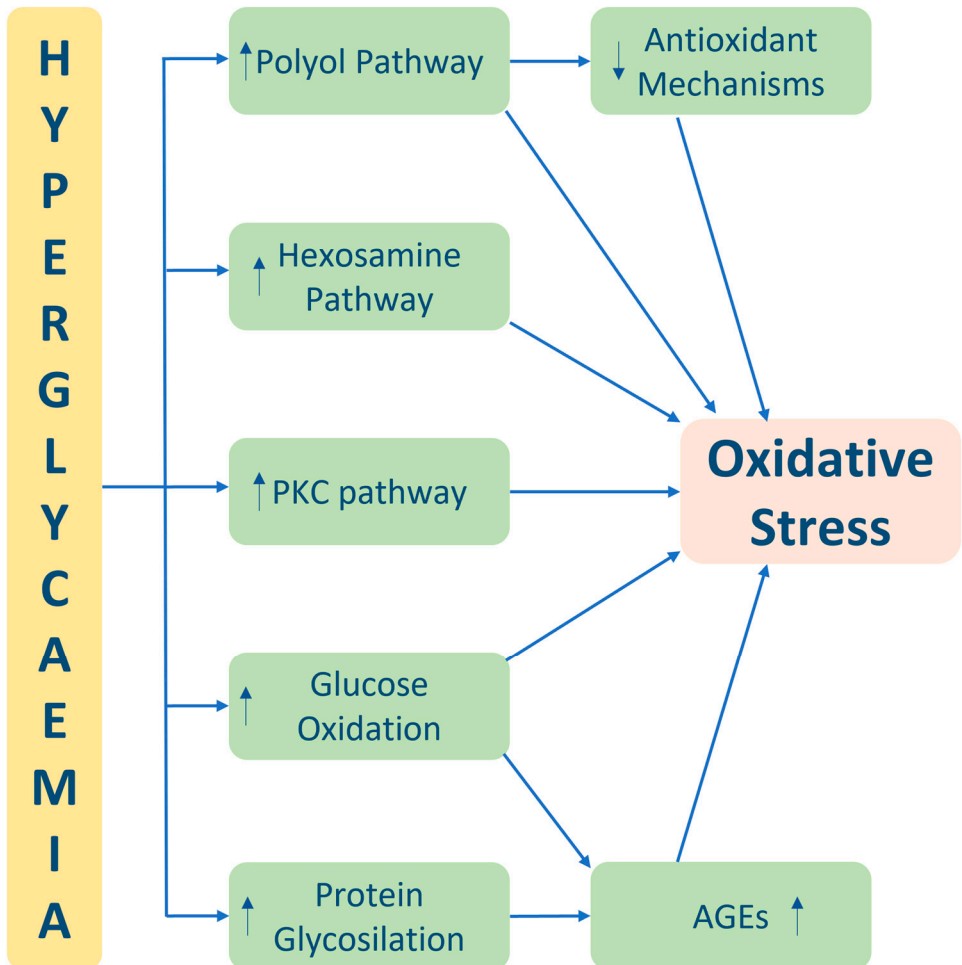

**Figure 1.** Main pathophysiological mechanism of hyperglycaemia induced oxidative stress. Abbreviations: PKC—Protein Kinase C and AGEs—advanced glycation end products.

## *2.1. Pentose Phosphate and Glycolytic Pathways and Oxidative Stress*

The primary cause of oxidative stress is undoubtedly the elevation in blood glucose concentration. In fact, once glucose enters cells, it undergoes oxidation through either the pentose phosphate pathway, leading to the production of biosynthetic molecules and NADPH, or through the glycolytic pathway [14]. Glycolysis continues with the Krebs cycle, resulting in the generation of nicotinamide adenine dinucleotide (NADH) and reduced flavin adenine dinucleotide (FADH2), which are utilised in oxidative phosphorylation to produce ATP. This process generates ROS such as hydrogen peroxide ($H_2O_2$), superoxide anion ($O2\bullet-$), and hydroxyl radicals ($\bullet OH$). Under normal physiological conditions, the antioxidant defence system comprising enzymes such as superoxide dismutase (SOD), catalase (CAT), and glutathione peroxidase (GPx) effectively neutralises ROS [15]. However, when blood glucose concentration becomes excessively high, the production of radicals is upregulated, leading to the inhibition of antioxidant systems. Consequently, this downregulation results in DNA damage and the subsequent production of DNA repair enzymes such as Poly-ADP Ribose Polymerase-1 (PARP-1), which inactivates glyceraldehyde-3-phosphate dehydrogenase (GAPDH), leading to the accumulation of glyceraldehyde-3-phosphate (GAP), Glucose 6-phosphate (G-6-P), and Fructose 6-phosphate (F-6-P). The elevation of these three intermediates contributes to various reactions that converge on oxidative stress: G-6-P and GAP can undergo autoxidation, leading to the formation of AGE precursors; G-6-P and F-6-P can follow the Polyol pathway, while GAP induces the activation of PKC [16]. Notably, with the increased glucose concentration, the enzyme hexokinase becomes saturated and cannot catalyse the formation of G-6-P. Consequently, glucose is

converted to sorbitol via aldose reductase, which is further converted to fructose by sorbitol dehydrogenase (SDH). This process consumes excess NADPH, which serves as a substrate for GPx to produce glutathione (GSH) [17]. Thus, the inhibition of antioxidant enzymes in this pathway also contributes to oxidative stress. Additionally, under hyperglycaemic conditions, SDH is upregulated, leading to increased fructose production, which is then converted into the triose-phosphates GAP and dihydroxyacetone-3-phosphate (DHA-3-P), ultimately resulting in the activation of PKC and oxidative stress [18].

Glucose in excess can undergo autoxidation to form glyoxal, while glucose-derived GAP and dihydroxyacetone-3-phosphate (DHAP) can undergo non-enzymatic dephosphorylation to form methylglyoxal. Both products, along with 3-deoxyglucosone (or Amadori product), serve as precursors for the formation of AGEs, which react with elements of the extracellular matrix (ECM), leading to AGE production. Specifically, the interaction of the carboxyl residue of a glucose molecule with the terminal ε-amino residue of a protein non-enzymatically results in the alteration of protein functionality. Once formed, AGEs interact with AGE receptors (RAGE), inducing oxidative stress and activating PKC, which upregulates NADPH oxidase and lipoxygenase, thereby generating ROS [19]. Moreover, the excess of GAP can be converted to dihydroxyacetone-3-phosphate (DHA-3-P), which is subsequently reduced to glycerol 3-phosphate. When combined with fatty acids, glycerol 3-phosphate forms diacylglycerol (DAG), capable of inducing PKC [20].

### 2.2. Inflammation and Oxidative Stress

There exists a strong correlation between inflammation and oxidative stress, as the immune system triggers the production of pro-inflammatory cytokines and chemokines, activating ROS-producing macrophages to eliminate pathogens. However, the chronic inflammatory state that arises in diabetes leads to continuous ROS production, resulting in cellular damage and the depletion of antioxidant systems [9]. In a reciprocal manner, ROS stimulate the expression of pro-inflammatory cytokines by activating transcription factors such as nuclear factor-kappa B (NF-κB) and activator protein-1 (AP-1). Additionally, the excess adipose tissue often observed in type II diabetes secretes pro-inflammatory cytokines, including tumour necrosis factor alpha (TNF-α) and interleukins 1 (IL-1) and 6 (IL-6), further amplifying oxidative stress [13]. Another mechanism implicated in diabetes complications is the hexosamine pathway. Hyperglycaemia leads to the accumulation of fructose, as mentioned earlier, which is converted to Glucosamine 6-phosphate and ultimately to UDP-N-Acetylhexosamine (UDP-GLCNac), potentiating O-Glucosamine-N-Acetyltransferase (OGT) activity [21]. OGT binds O-GlcNAc to serine and threonine residues of transcription factors, such as Sp1, thereby altering gene expression. Sp1, a transcription factor commonly implicated in diabetic complications, regulates the expression of various genes, including tissue-type plasminogen activator inhibitor-1 (PAI-1) and transforming growth factor-β1 (TGF-β1). PAI-1 is believed to play a role in diabetic neuropathy, although further studies are required to fully elucidate its function [22], as it likely impairs fibrinolysis in neural blood vessels, promoting nerve ischemia and oxidative stress. Conversely, the upregulation of TGF-β1 induces ROS production in vascular smooth muscle and endothelial cells by activating NADPH oxidase. TGF-β1 is involved in diabetic nephropathy, stimulating collagen formation and inhibiting mesangial cell mitosis [8,14]. Furthermore, the accumulation of ROS activates mitochondrial uncoupling protein-2 (UCP-2), reducing ATP production and initiating a cascade of events that ultimately impairs insulin secretion by pancreatic beta cells [23]. Moreover, oxidative stress induced by hyperglycaemia has been shown to inhibit the expression of the insulin gene and promote beta-cell apoptosis. These conditions collectively disrupt beta-cell function and insulin release, contributing to hyperglycaemia and exacerbating oxidative stress [24].

## 3. Oxidative Stress Role in the Development of Type 2 Diabetes Complications

In diabetic pathology, which is becoming progressively more prevalent worldwide [2], inflammation is playing an increasingly significant role. Navarro and Mora [25], building

upon the hypothesis proposed by Pickup and Crook [26], suggest that diabetes is transitioning from a metabolic disorder to a genuine inflammatory pathology, where prolonged and improper activation of the immune system contributes to the development and progression of the disease. In addition to its involvement in the onset of diabetes [27–29], an inflammatory state plays a crucial role in the advancement of the disease and the development of both microvascular and macrovascular complications [20,30–34]. Chronic hyperglycaemia is the primary driver of this process, as it induces a persistent activation of the innate immune system and triggers oxidative stress, leading to the production of harmful free radicals [35,36] that adversely affect the body [37–40]. Free radicals, specifically ROS and reactive nitrogen species (RNS), are highly unstable molecules with unpaired electrons, making them potent oxidants. Although our body naturally produces these reactive species, excessive production, as observed in diabetes, is detrimental [41]. Many cells in our body, such as red blood cells and endothelial cells, are particularly vulnerable to free radicals due to their high levels of polyunsaturated fatty acids, molecular oxygen, and ferrous ions [42,43]. In diabetic pathology, chronic inflammation and oxidative stress create a vicious cycle: each stimulates the other, resulting in mutual amplification. ROS and RNS, produced as a consequence of inflammation, can promote the transcription of growth factors like Nf-Kb and AP-1, which in turn stimulate the production of inflammatory proteins [44].

Free radicals play a crucial role in the onset and progression of diabetic complications through three different pathways: the aldose reductase pathway, the PKC pathway, and the production of AGEs. The aldose reductase pathway normally converts glucose to sorbitol through the simultaneous oxidation of sorbitol to fructose in a nicotinamide adenine dinucleotide phosphate (NADPH)-dependent manner. In diabetes, chronic hyperglycaemia saturates this pathway early on, causing more than 30% of glucose to be metabolised through the polyol pathway [16,45]. The increased utilisation of this pathway depletes intracellular NADPH and increases extracellular NADH levels. The excess NADH serves as a substrate for the enzyme NADH oxidase, leading to the generation of excessive ROS [46]. This mechanism has been primarily implicated in the development of diabetic retinopathy due to sorbitol accumulation in the retina [47–49].

Excess glucose in the body can also react spontaneously with the amino groups of plasma proteins, forming Schiff bases that subsequently contribute to the formation of AGEs [50]. These AGEs react with other membrane molecules in endothelial cells and blood vessels, leading to structural and functional alterations in proteins and the activation of pro-inflammatory gene transcription [31].

Chronic hyperglycaemia induces oxidative stress through the PKC-dependent activation of the NADPH oxidase pathway. While this enzyme is normally present primarily in phagocytic cells, in diabetes, it is also expressed in fibroblasts, endothelial cells, and smooth muscle cells, where it becomes the main producer of ROS. PKC also activates endothelial nitric oxide synthase (eNOS), NADPH oxidase, phospholipase A2 (PLA2), endothelin-1 (ET-1), vascular endothelial growth factor-B (TGF-B), and NF-KB, all of which play a role in the development of complications [51]. PKC, when activated by diacylglycerol, also directly transcribes genes involved in protein synthesis, leading to direct cell damage, capillary occlusion, and reduced blood flow [52,53].

During periods of hyperglycaemia, the cytokine cascade resulting from the interaction between ROS and inflammation is characterised by an increase in monocyte chemoattractant protein-1 and a decrease in insulin-like growth factor-1 levels [54–56]. These proteins, as observed by Al Hannan et al. [57], can induce dedifferentiation of adipose tissue through macrophages, resulting in increased hyperinsulinemia and insulin resistance. It is also possible that, in this cellular crosstalk, a role can also be played by the opioid system, though more studies are needed [58]. Our body normally possesses various protection systems against ROS and oxidative stress, with the main ones being SOD, catalase, and glutathione peroxidase. However, in diabetes, these defence mechanisms are compro-

mised due to chronic inflammation, contributing to the progression of damage and the disease itself.

Microvascular complications of diabetes include retinopathy, nephropathy, and neuropathy. Retinopathy, in particular, mainly develops due to damage to small retinal vessels and connective tissue, sometimes resulting in the formation of small haemorrhages. The retinal tissue is highly susceptible to damage mediated by oxidative stress due to its high concentration of polyunsaturated fats [59]. Diabetic nephropathy involves the interstitial and glomerular membranes, which are damaged by inflammatory cytokines and ROS, leading to the loss of proteins such as albumin and a decreased glomerular filtration rate [60,61]. Additionally, changes in kidney haemodynamics may occur due to thinning of the basement membrane, expansion of renal mesangial cells, and hyperplasia of the extracellular matrix. Increased ROS levels in the kidney cause vasoconstriction, endothelial dysfunction, and enhanced sodium reabsorption [47]. Diabetic neuropathy, the most common microvascular complication of diabetes, affects approximately 50% of patients after 20 years of the disease. Oxidative stress in the central nervous system can induce neuronal apoptosis [62] and impair the ability to repair and regenerate neurons effectively.

## 4. Dietary Strategies to Counteract Oxidative Stress

Oxidative stress is a major contributing factor in the complications of type II diabetes (DMT2), prompting scientists to investigate the most effective diets to counteract oxidative stress. Notably, antioxidants such as glutathione, coenzyme Q10, and α-lipoic acid have been found to restore insulin sensitivity in DMT2 [63].

The Mediterranean diet (MD) is widely considered the most effective diet for preventing non-communicable diseases like cardiovascular diseases (CVD) and DMT2 [64]. The MD emphasises the consumption of vegetables, fruits (1–2 portions a day), whole grains, extra virgin olive oil, legumes, and nuts [65]. It also includes moderate amounts of fish and dairy products while minimising the intake of meat, processed, and industrial foods [66].

The MD's efficacy against oxidative stress is attributed to the presence of natural antioxidants found primarily in vegetables, fruits, and spices. These foods contain phytochemicals such as polyphenols, phytosterols, and carotenoids, which activate the nuclear factor erythroid 2-related factor 2 (NRF2), a master regulator gene that activates genes involved in combating oxidative stress and facilitating detoxification (Figure 2) [67].

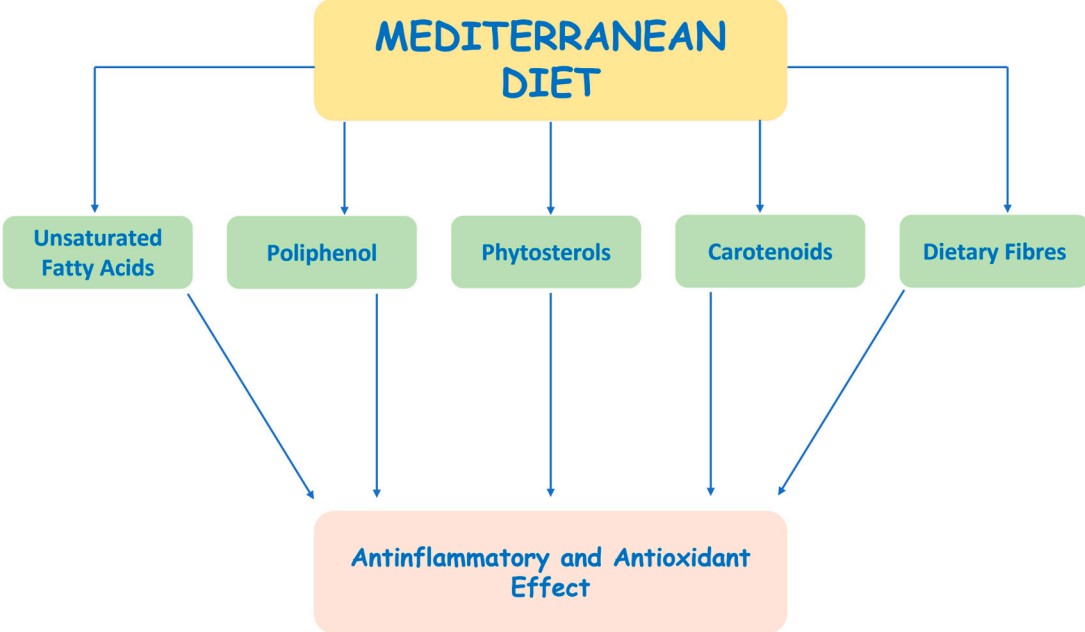

**Figure 2.** Mediterranean diet nutrients that exert an anti-inflammatory and antioxidant effect.

Polyphenols, abundant in MD foods, exhibit strong antioxidant and anti-inflammatory properties and have been extensively studied. For instance, quercetin-3-glucoside from apples and onions reduces oxidative stress similar to metformin, while epigallocatechin-3-O-gallate (EGCG) found in green tea increases energy expenditure and decreases inflammation and oxidative stress by inhibiting the NF-kB pathway [68]. Resveratrol, present in red wine, grapes, peanuts, and plums, has hormetic properties that enhance neural cells' ability to cope with stress and promote optimal function and longevity [69]. Resveratrol also activates SIRT1, improving insulin sensitivity, and inhibits NF-kB, as do other polyphenols [70]. Other sources of polyphenols include rosemary (carnosic and rosmarinic acids) and extra virgin olive oil (oleuropein and hydroxytyrosol) [68].

Carotenoids, precursors of vitamin A, not only stimulate the immune system but also contribute to the elimination of ROS by activating NRF2. Lycopene, an extract from tomatoes, has been shown to improve DMT2 by activating antioxidant systems such as SOD and glutathione peroxidase (GPx) [71]. Fresh fruits and vegetables are also rich in vitamin C, a potent essential antioxidant, while green leafy vegetables and dried fruits provide high levels of vitamin E, which activates NRF2, heat shock proteins, and protects cell membranes from ROS while downregulating NF-kB [66].

The MD's inclusion of extra virgin olive oil (EVOO) provides a significant amount of oleic acid, while fish and nuts are rich in omega-3 polyunsaturated fatty acids (alpha-linolenic acid). Oleic and linoleic acids have anti-inflammatory effects by reducing pro-inflammatory eicosanoids derived from arachidonic acid and increasing the production of anti-inflammatory eicosanoids [72].

Fibre-rich foods, such as whole grains, legumes, dried fruits, and vegetables, are a notable feature of the MD. Dietary fibre not only reduces post-prandial blood sugar levels but also enables the synthesis of short-chain fatty acids (SCFA) by the gut microbiota, which possess anti-inflammatory properties. Furthermore, fibre enhances the diversity of the microbiome, which plays a crucial role in the immune system's anti-inflammatory response [73].

Considering the antioxidant and anti-inflammatory qualities that make the MD suitable for combating oxidative stress induced by hyperglycaemia, strategies like fasting and calorie restriction (CR) should be considered. Intermittent fasting (IF) involves a 60% or more energy restriction for 2–3 days a week or alternate-day fasting, while time-restricted feeding (TRF) limits the daily eating window to 8–10 h [74]. IF and TRF exhibit similar effects, counteracting oxidative stress and insulin resistance [75].

These strategies have been shown to reduce inflammation, as evidenced by decreased levels of tumour necrosis factor-alpha (TNF-alpha). IF and TRF also alleviate oxidative stress in brain tissue by enhancing the functionality of antioxidant systems and increasing uric acid levels [76,77].

One significant finding regarding CR is its ability to counteract the AKT activation pathway, which leads to mTOR activation and cell proliferation. Fasting and CR exert their beneficial effects through the activation of adenosine monophosphate-activated protein kinase (AMPK), which induces autophagy, promotes mitochondrial biogenesis, and combats oxidative stress and inflammation by degrading and recycling cellular components [78].

Sirtuin 3 (SIRT3), a mitochondrial deacetylase, plays a crucial role in these strategies [79]. SIRT3 controls enzymes involved in oxidative phosphorylation, promotes DNA repair, and activates antioxidant systems to protect cellular components from ROS damage [80,81]. Studies on mouse models have revealed the significance of SIRT3 in diabetes, as it regulates glucose uptake and glycolysis in endothelial cells [82]. Additionally, during CR, SIRT3 expression increases in various tissues, including skeletal and cardiac muscles, as well as white and brown adipose tissues, while genetically obese mice show reduced SIRT3 expression and decreased mitochondrial function in brown adipose tissue [83,84]. This heightened SIRT3 expression is particularly noteworthy as it facilitates the activation of acetyl-CoA synthase, an enzyme responsible for converting acetate to acetyl-CoA, an essential step during extended fasting [83,84]. In contrast, a persistent high-fat diet led

to the inhibition of SIRT3 activity and an increase in overall acetylation of mitochondrial proteins [85].

After 6–8 h of fasting, the body produces β-hydroxybutyrate, a ketone body that reduces inflammation and oxidative stress by increasing antioxidant enzyme activity and improving mitochondrial function [86,87]. The production of ketone bodies is a characteristic shared with the ketogenic diet, which restricts carbohydrate intake to a maximum of 50 g per day. However, the therapeutic use of fasting and ketogenic diets in drug-compensated diabetes is still being investigated due to the potential risk of severe hypoglycaemia [88].

Another essential aspect in preventing oxidative stress and metabolic diseases like DMT2 is chrononutrition [89]. Chrononutrition is based on the theory of a biological clock located in the hypothalamus, which regulates hormone production in response to day–night cycles perceived by retinal photoreceptors. The timing of food consumption is crucial, as catabolic hormones such as adrenaline are primarily produced in the first part of the day, while anabolic hormones like growth hormone (GH) are predominant in the afternoon and evening. Excessive food intake during the latter hours is more likely to be stored as fat. Therefore, it is advisable to consume higher-calorie foods with higher sugar content, such as complex carbohydrates, vegetables, and legumes, during the earlier part of the day. In the evening, foods with a higher protein content are preferable because they are utilised as substrates for protein synthesis and muscle mass accumulation due to the rise in GH levels [90].

In conclusion, adopting a balanced lifestyle is crucial for preventing metabolic diseases and delaying cellular ageing. It is important to consume a varied diet that incorporates a wide range of foods to ensure the intake of all essential nutrients necessary for good health [91].

## 5. Physical Exercise and Oxidative Stress

Physical exercise is widely recommended for its numerous health benefits, which include improvements in cardiovascular and respiratory function, body composition, and glycaemic control [92]. Recent studies have revealed that physical exercise stimulates the production of ROS due to increased activity in skeletal muscles, leading to a temporary imbalance known as "exercise-induced oxidative stress" [93–95]. A significant source of ROS production during physical exercise is represented by the NOX2 enzyme complex, which is present in the sarcolemma membrane, transverse tubule, and sarcoplasmic reticulum of skeletal muscle [95]. Muscle fibre depolarisation triggers the release of ATP via pannexin, which activates purinergic receptors linked to the IP3 signalling pathway, resulting in calcium release from the sarcoplasmic reticulum and activation of PKC. Consequently, PKC phosphorylates the NOX2 cytoplasmic subunit (p47/Phox), causing a conformational change that enables its association with the NOX2 plasma membrane subunits, thus restoring the functionality of the NOX2 complex [96]. In addition, skeletal muscle ROS production is influenced by various mechanisms, including the activation of enzymes such as neural nitric oxide synthase (nNOS), PLA2, and, to a lesser extent, the generation of $O_2\bullet^-$ from the mitochondrial electron transport chain [95]. Furthermore, the endothelium of blood vessels that supply muscle tissue also contributes to ROS production, with xanthine oxidase and eNOS playing a significant role [95]. The production of ROS during exercise is transient, with a peak of biomarker levels of oxidative damage between 24 and 48 h [97], inducing adaptation following the "hormesis" principle [93], thus enabling the activation of various signalling pathways that converge on the activation of transcription factors like Nrf2, NF-κB, and the cofactor PGC-1α [94]. To elaborate, Nrf2 is held in the cytosol by Keap1 under physiological conditions. However, after oxidative stimulation, the thiol groups of Keap1 undergo covalent modifications, causing its dissociation from Nrf2, which in turn translocates into the nucleus and promotes antioxidant enzyme expression by binding to Antioxidant Response Elements [98,99]. In the case of the NF-κB complex, ROS disrupts the trimeric complex formed by the inhibitory protein IκB (NF-κB inhibitor) and the p50/p65 protein dimer. Concurrently, ROS induces redox changes that lead to the

phosphorylation of the IκB subunit, activating its proteolytic degradation. This results in the release of the IκB subunit from the p60/p65 heterodimer, allowing NF-κB to translocate into the nucleus and initiate the transcription of pro-inflammatory cytokines such as IL-1, IL-6, and TNFα [98,99]. PGC-1α acts as a transcriptional coactivator by recruiting and co-regulating multiple transcription factors that regulate skeletal muscle gene expression, including Nrf-2, thereby promoting mitochondrial biogenesis and increased expression of antioxidant enzymes [98,99]. The activation of Nrf2, NF-κB, and PGC-1α through exercise leads to an increase in the content and activity of enzymes such as SOD, CAT, and GPx [94,96,97,99]. Furthermore, it enhances glutathione levels in tissues, elevates plasma antioxidant capacity [97], and promotes mitochondrial biogenesis [100], thereby improving antioxidant defence and reducing levels of biomarkers of oxidative stress (Figure 3) [101].

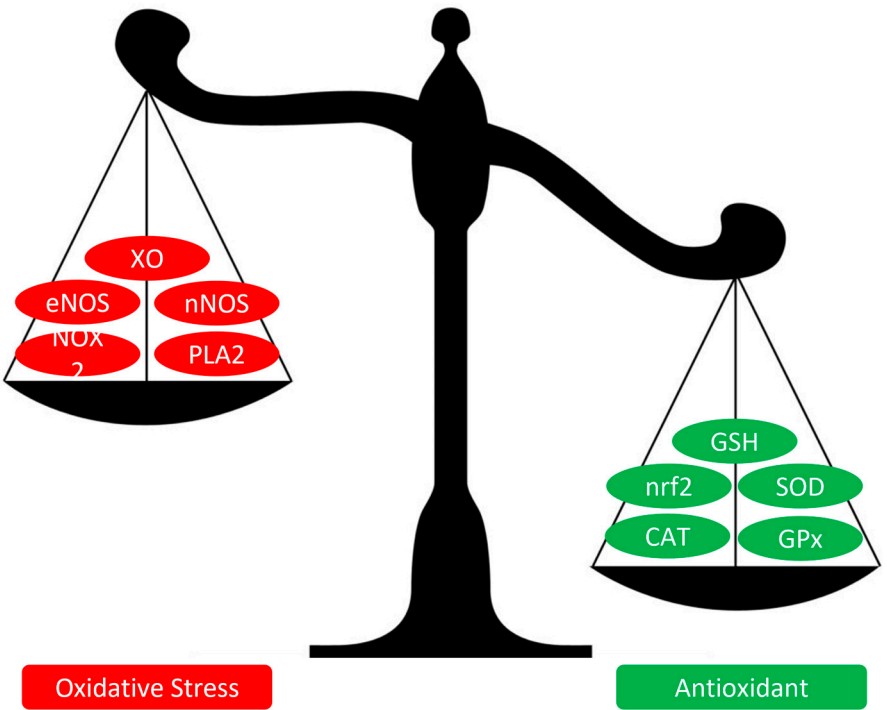

**Figure 3.** Oxidative stress and antioxidant imbalance due to physical exercise. Abbreviations: XO—xanthine oxidase, eNOS—endothelial nitric oxide synthase, nNOS—neural nitric oxide synthase, NOX2—NADPH oxidase-2, PLA2—phospholipase A2, GSH—glutathione, nrf2—nuclear factor erythroid 2-related factor 2, SOD—superoxide dismutase, CAT—catalase, and GPx—glutathione peroxidase.

### 5.1. Effects of Different Physical Exercise Protocols on Oxidative Stress Markers in Patients with Type 2 Diabetes Mellitus

Various physical exercise protocols have been recommended for patients with T2DM, including continuous moderate-intensity exercise (CMIE) [102], resistance exercise (RE) [103], high-intensity interval exercise (HIIE) [100], and concurrent exercise (CE), which combines CMIE and RE [104].

### 5.2. Resistance Exercise

RE involves performing both monoarticular or polyarticular movements against resistance and returning to the start position [103]. While RE has been shown to improve metabolic health in T2DM patients [105], its impact on oxidative stress parameters in this population remains inconclusive [106]. Remarkably, the response to RE on oxidative stress markers seems to vary depending on the health status of the patients. In healthy individuals, RE protocols involving three sets at 65–70% of one repetition maximum (RM) performed three times a week for 6–8 weeks resulted in decreased plasma levels of malondi-

aldehyde and increased blood activity of GPx [107,108]. Similarly, high-intensity protocols ($3 \times 3$–6 repetitions at 85–90% of 1 RM) have been associated with an increase in blood SOD activity [107]. However, studies on healthy subjects [107,108] involved younger participants under the age of 30 years, compared to the trial enrolling diabetic patients [106], whose participants were over 50 years old. Due to this, it is worth noting that the effectiveness of T2DM might have been influenced by age-related factors or lower adaptive responses typical of this patient group.

### 5.3. Continuous Moderate-Intensity Exercise

CMIE, or "aerobic exercise", involves cyclical modalities such as walking, jogging, and cycling, engaging large muscle groups [109]. Studies examining the effects of CMIE on oxidative stress biomarkers in T2DM patients generally show improvements in redox balance due to an increase in blood antioxidant biomarkers [107,110–112] and a decrease in blood protein oxidation biomarkers [112] and DNA in urine [113]. Moreover, these positive responses align with improvements in clinical parameters such as increased cardiovascular function [110–112], lipid profile control [107], body composition enhancement [111,113], and glycaemic control [111,113]. Of note, only one study reported no significant changes in antioxidant abundance/activity or markers of oxidative stress damage, specifically malondialdehyde levels, even though positive effects on clinical parameters like fasting glycaemia, HOMA index, and body fat percentage were confirmed [100]. Krause et al. noted a significant increase in blood CAT activity following a moderate-intensity free-walking protocol, along with a decrease in carbonylated protein levels with low/moderate-intensity protocols. Though reporting an approximately 2% body fat percentage reduction, this was not statistically significant, and there were no significant changes in fasting glycaemia and the HOMA index [112]. The response to CMIE on oxidative stress levels appears to depend on the intervention's duration and the subjects' age. It has been shown that participants over 60 years of age may require an intervention period of 16 weeks or more, while those below 60 years of age may experience changes with 12-week interventions, with an exercise duration equal to or greater than 30 min at moderate intensity [114].

### 5.4. High-Intensity Interval Exercise

HIIT involves the repeated performance of short bursts of intense exercise (ranging from 10 s to 4 min) at an intensity level exceeding the anaerobic threshold, followed by periods of recovery at low intensity or complete rest [115]. Limited evidence exists regarding the effects of HIIT on oxidative stress markers in patients with T2DM. Mitranun et al. [100] conducted a clinical trial enrolling 43 patients randomly assigned to three groups: HIIT, CMIE, and a control group with no exercise intervention. The HIIT group participated in 12 weeks of treadmill jogging with three weekly sessions, with a gradually increasing session duration from 30 to 60 min. Although no statistically significant changes in SOD concentrations were observed in any of the experimental groups, the HIIT group exhibited a significant decrease in malondialdehyde levels and an increase in GPx enzyme activity. These results correlated with a reduced percentage of HbA1c, fasting glycaemia, and HOMA index, as well as improvements in other clinical parameters related to cardiovascular function, lipid profile, and body composition [100].

### 5.5. Concurrent Exercise

The CE protocol typically combines CMIE with RE within the same training session [116]. CE has shown benefits in glycaemic control (both fasting glycaemia and HbA1c levels) and body composition in patients with T2DM, and it has been found to induce positive effects on oxidative stress parameters [117]. In this context, CE has been associated with enhancements in blood antioxidant biomarkers like GSH and SOD, along with a decrease in malondialdehyde levels [106]. These positive responses have been reported to be observed after 8–16 weeks of intervention [100], thus highlighting the potential of CE protocol in promoting a more favourable redox balance in T2DM.

## 6. Conclusions

The pathogenesis of oxidative stress remains complex and multifaceted, giving rise to uncertainties in our understanding of its exact mechanisms. While it is widely accepted that increased production of ROS and compromised antioxidant defences contribute to oxidative stress, the precise triggers and molecular pathways involved are still under investigation. Various factors, such as chronic hyperglycaemia, mitochondrial dysfunction, inflammation, and altered signalling pathways, interact and influence ROS generation and antioxidant capacity in a dynamic and interdependent manner. Additionally, the role of oxidative stress as a cause or consequence of diabetic complications is still a matter of ongoing research [118–121]. Elucidating the intricate network of interactions and unravelling the specific molecular events driving oxidative stress will be crucial for developing targeted therapeutic approaches and improving our understanding of diabetes pathogenesis as a whole.

**Author Contributions:** Conceptualisation, A.C. and M.D.; writing—original draft preparation, A.C., M.D. and A.M.; writing—review and editing, A.C., M.D., V.R., R.G., L.R., E.V. and A.M.; supervision T.S., M.P.M., A.G., R.M., M.M. and F.C.S. All authors have read and agreed to the published version of the manuscript.

**Funding:** The authors received no financial support for the research, authorship, and/or publication of this article.

**Institutional Review Board Statement:** The authors have reviewed literature data and have reported results coming from studies approved by local ethics committees.

**Informed Consent Statement:** Not applicable.

**Data Availability Statement:** No dataset was generated for the publication of this article.

**Acknowledgments:** We wish to thank Francesca Dello Iacovo for the English revision of the manuscript.

**Conflicts of Interest:** The authors declare no conflict of interest.

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
