# Peer review of "Oxidative Stress in Type 2 Diabetes: Impacts from Pathogenesis to Lifestyle Modifications"

_cimb, doi:10.3390/cimb45080420_

Round 1
Reviewer 1 Report
Comments and Suggestions for Authors
Congratulations on your review. I do not have much to comment. The manuscript is adequate. If you could add some more information about molecule SIRT3 and some more data from experiments I would be very pleased to read something more detailed.
The manuscript addresses one of the hottest topics in current literature. The truth is that because of the interest around this topic the are a lot of similar manuscripts. However this is a well-written and detailed one that I read with geat pleasure. I did not found any plagiarism issues. Some more refereces: 1.Sirtuin 1 and sirtuin 3: physiological modulators of metabolism DOI: 10.1152/physrev.00022.2011 2.Sirtuin Evolution at the Dawn of Animal Life. DOI: 10.1093/molbev/msac192 3.Aging Hallmarks and the Role of Oxidative Stress DOI: 10.3390/antiox12030651
I would be very satisfied if authors could provide more information about molecule SIRT3 (maybe some more specific data)
Author Response
Reviewer 1
Congratulations on your review. I do not have much to comment. The manuscript is adequate. If you could add some more information about molecule SIRT3 and some more data from experiments I would be very pleased to read something more detailed.
The manuscript addresses one of the hottest topics in current literature. The truth is that because of the interest around this topic the are a lot of similar manuscripts. However this is a well-written and detailed one that I read with geat pleasure. I did not found any plagiarism issues. Some more refereces: 1.Sirtuin 1 and sirtuin 3: physiological modulators of metabolism DOI: 10.1152/physrev.00022.2011 2.Sirtuin Evolution at the Dawn of Animal Life. DOI: 10.1093/molbev/msac192 3.Aging Hallmarks and the Role of Oxidative Stress DOI: 10.3390/antiox12030651
I would be very satisfied if authors could provide more information about molecule SIRT3 (maybe some more specific data).
R. We express our sincere gratitude to the esteemed reviewer for his/her invaluable comments, which have undeniably elevated the quality of our manuscript. Consequently, we have diligently implemented the requisite revisions to address your insightful suggestions. Supplementary data has been seamlessly integrated into lines 59-67 and 308-316, and the suggested references have been thoughtfully incorporated. It is our considered decision not to include the reference "Sirtuin Evolution at the Dawn of Animal Life," as we hold the view that it diverges significantly from the central theme and scope of our manuscript.
Reviewer 2 Report
Comments and Suggestions for Authors
This thematic review addresses the role of oxidative stress in the development of diabetes and its related complications. The role of oxidative stress has been an important piece of the puzzle for the understanding of the complex mechanism by which diabetes and its complications are developed. Many researchers had shown that changes in lifestyle such as unhealthy diet and physical inactivity are strongly associated with the growing prevalence of this disease.
In general, it is well written manuscript that summarizes the current knowledge of the mechanisms by which oxidative stress contributes on the progression of the pathogenesis of diabetes. Although, this has been a topic of debate in many previous reviews this one added another twist to the problem, the discussion of how lifestyle modifications such us dietary changes (the use of the Mediterranean diet) and exercise can help reduce oxidative stress in diabetes.
Author Response
Reviewer 2
This thematic review addresses the role of oxidative stress in the development of diabetes and its related complications. The role of oxidative stress has been an important piece of the puzzle for the understanding of the complex mechanism by which diabetes and its complications are developed. Many researchers had shown that changes in lifestyle such as unhealthy diet and physical inactivity are strongly associated with the growing prevalence of this disease.
In general, it is well written manuscript that summarizes the current knowledge of the mechanisms by which oxidative stress contributes on the progression of the pathogenesis of diabetes. Although, this has been a topic of debate in many previous reviews this one added another twist to the problem, the discussion of how lifestyle modifications such us dietary changes (the use of the Mediterranean diet) and exercise can help reduce oxidative stress in diabetes.
R. We extend our heartfelt gratitude to the esteemed reviewer for his/her invaluable comments and appreciation.